# A two-lane mechanism for selective biological ammonium transport

Gordon Williamson[1†], Giulia Tamburrino[2,3†], Adriana Bizior[1†], Mélanie Boeckstaens[4†], Gaëtan Dias Mirandela[1‡], Marcus G Bage[2,3], Andrei Pisliakov[2,3], Callum M Ives[2], Eilidh Terras[1], Paul A Hoskisson[1], Anna Maria Marini[4], Ulrich Zachariae[2,3]*, Arnaud Javelle[1]*

[1]Strathclyde Institute of Pharmacy and Biomedical Sciences, University of Strathclyde, Glasgow, United Kingdom; [2]Computational Biology, School of Life Sciences, University of Dundee, Dundee, United Kingdom; [3]Physics, School of Science and Engineering, University of Dundee, Dundee, United Kingdom; [4]Biology of Membrane Transport Laboratory, Department of Molecular Biology, Université Libre de Bruxelles, Gosselies, Belgium

**Abstract** The transport of charged molecules across biological membranes faces the dual problem of accommodating charges in a highly hydrophobic environment while maintaining selective substrate translocation. This has been the subject of a particular controversy for the exchange of ammonium across cellular membranes, an essential process in all domains of life. Ammonium transport is mediated by the ubiquitous Amt/Mep/Rh transporters that includes the human Rhesus factors. Here, using a combination of electrophysiology, yeast functional complementation and extended molecular dynamics simulations, we reveal a unique two-lane pathway for electrogenic $NH_4^+$ transport in two archetypal members of the family, the transporters AmtB from *Escherichia coli* and Rh50 from *Nitrosomonas europaea*. The pathway underpins a mechanism by which charged $H^+$ and neutral $NH_3$ are carried separately across the membrane after $NH_4^+$ deprotonation. This mechanism defines a new principle of achieving transport selectivity against competing ions in a biological transport process.

*For correspondence:
u.zachariae@dundee.ac.uk (UZ);
arnaud.javelle@strath.ac.uk (AJ)

†These authors contributed equally to this work

Present address: ‡Wellcome Trust Centre for Cell Biology, Michael Swann Building, The King's Buildings, University of Edinburgh, Edinburgh, United Kingdom

Competing interests: The authors declare that no competing interests exist.

## Introduction

The transport of ammonium across cell membranes is a fundamental biological process in all domains of life. Ammonium exchange is mediated by the ubiquitous ammonium transporter/methyl-ammonium-ammonium permease/Rhesus (Amt/Mep/Rh) protein family. The major role of bacterial, fungal, and plant Amt/Mep proteins is to scavenge ammonium for biosynthetic assimilation, whereas mammals are thought to produce Rh proteins in erythrocytes, kidney, and liver cells for detoxification purposes and to maintain pH homeostasis (*Biver et al., 2008*; *Huang and Ye, 2010*). In humans, Rh mutations are linked to pathologies that include inherited hemolytic anemia, stomatocytosis, and early-onset depressive disorder (*Huang and Ye, 2010*). Despite this key general and biomedical importance, so far, no consensus on the pathway and mechanism of biological ammonium transport has been reached.

High-resolution structures available for several Amt, Mep and Rh proteins show a strongly hydro-phobic pore leading towards the cytoplasm (*Andrade et al., 2005*; *Gruswitz et al., 2010*; *Khademi et al., 2004*; *Lupo et al., 2007*; *van den Berg et al., 2016*). This observation led to the conclusion that the species translocated through Amt/Mep/Rh proteins is neutral $NH_3$. However, this view has been experimentally challenged, first for some plant Amt proteins (*Ludewig et al., 2002*; *Mayer et al., 2006*; *McDonald and Ward, 2016*; *Neuhäuser et al., 2014*), followed by further in-vitro studies revealing that the activity of bacterial Amt proteins is electrogenic (*Mirandela et al.,*

*2019*; *Wacker et al., 2014*). Taken together, these findings renewed a long-standing debate on the mechanism by which a charged molecule is translocated through a hydrophobic pore and how selectivity for $NH_4^+$ over competing ions is achieved.

Here, we reveal the pathways, mechanism, and key determinants of selectivity of electrogenic ammonium transport in Amt and Rh proteins, unifying the diverse observations that led to these seemingly incompatible suggestions. The transport mechanism is underpinned by the separate transfer of $H^+$ and $NH_3$ on a unique two-lane pathway following $NH_4^+$ sequestration and deprotonation. This mechanism ensures that ammonium – which occurs mainly in protonated form in the aqueous phase – is efficiently translocated across the membrane, while maintaining strict selectivity against $K^+$, a monovalent cation of similar size. This previously unobserved principle is likely to form a new paradigm for the electrogenic members of the Amt/Mep/Rh family. Similar mechanisms may be utilized by other membrane transporters to facilitate the selective translocation of pH-sensitive molecules.

## Results and discussion

### AmtB and NeRh50 activity is electrogenic

Motivated by our finding that the activity of *Escherichia coli* AmtB is electrogenic (*Mirandela et al., 2019*), we first investigated the transport mechanism of the Rh50 protein from *Nitrosomonas europaea* (NeRh50). Rh and Amt proteins are distant homologs, and thus a functional distinction between both subfamilies has been proposed (*Huang and Ye, 2010*). The architecture of NeRh50 is highly representative of Rh proteins (*Gruswitz et al., 2010*; *Lupo et al., 2007*) which have been repeatedly reported to serve as electroneutral $NH_3$ or $CO_2$ gas channels (*Cherif-Zahar et al., 2007*; *Hub et al., 2010a*; *Li et al., 2007*; *Lupo et al., 2007*; *Weidinger et al., 2007*). The activity of purified NeRh50 reconstituted into liposomes was quantified using Solid-Supported Membrane Electrophysiology (SSME) (*Bazzone et al., 2017*) experiments, where we recorded a $NH_4^+$-selective current (*Figure 1*) with a decay rate that is strongly dependent on the lipid-to-protein ratio (LPR; *Table 1*, *Figure 1—figure supplement 1*). Expressed in a *Saccharomyces cerevisiae* triple-mepΔ strain, deprived of its three endogenous Mep ammonium transporters, NeRh50 complemented the growth defect on minimal medium containing ammonium as sole nitrogen source (*Figure 1*). The electrogenic transport activity observed for NeRh50 and AmtB may suggest a common transport mechanism amongst the distant Amt and Rh proteins, but more experiments are needed to conclusively confirm this. Also, the $NH_4^+$ selectivity of both transporters further highlighted the question of how these proteins achieve selective charge translocation through their hydrophobic pore.

### Two interconnected water wires form an $H^+$ translocation pathway in AmtB

We next made use of the most substantive body of structural information available for the archetypal ammonium transporter AmtB from *E. coli* and its variants to decipher the molecular mechanism of electrogenic $NH_4^+$ transport (*Dias Mirandela et al., 2018*). Computational (*Wang et al., 2012*) and experimental studies (*Ariz et al., 2018*) have suggested that deprotonation of $NH_4^+$ is likely to be a major step in ammonium transport. We therefore aimed to identify dynamic polar networks across AmtB that could form a transfer pathway through the protein for the translocation of $H^+$, coming from $NH_4^+$ deprotonation. AmtB forms homotrimers in the cytoplasmic membrane, in which each monomer exhibits a potential periplasmic $NH_4^+$ binding region (S1) near residue D160, followed by a strongly hydrophobic pore leading towards the cytoplasm (*Figure 2A*; *Khademi et al., 2004*). Two highly conserved histidine residues, H168 and H318, protrude into the lumen, forming the family's characteristic 'twin-His' motif (*Javelle et al., 2006*). The only variation in the twin-His motif in members of the Amt/Mep/Rh family is in numerous fungal Mep transporters where the first His, corresponding to H168, is replaced by a Glu (*Javelle et al., 2006*). The general conservation pattern in the AmtB pore, as analysed with ConSurf (*Ashkenazy et al., 2016*), is shown in *Figure 2—figure supplement 1*.

To locate potential polar transfer routes, we performed atomistic molecular dynamics (MD) simulations of AmtB in mixed lipid bilayers. The simulations initially showed hydration of part of the putative hydrophobic $NH_3$ pathway from the twin-His motif to the cytoplasm (cytoplasmic water wire –

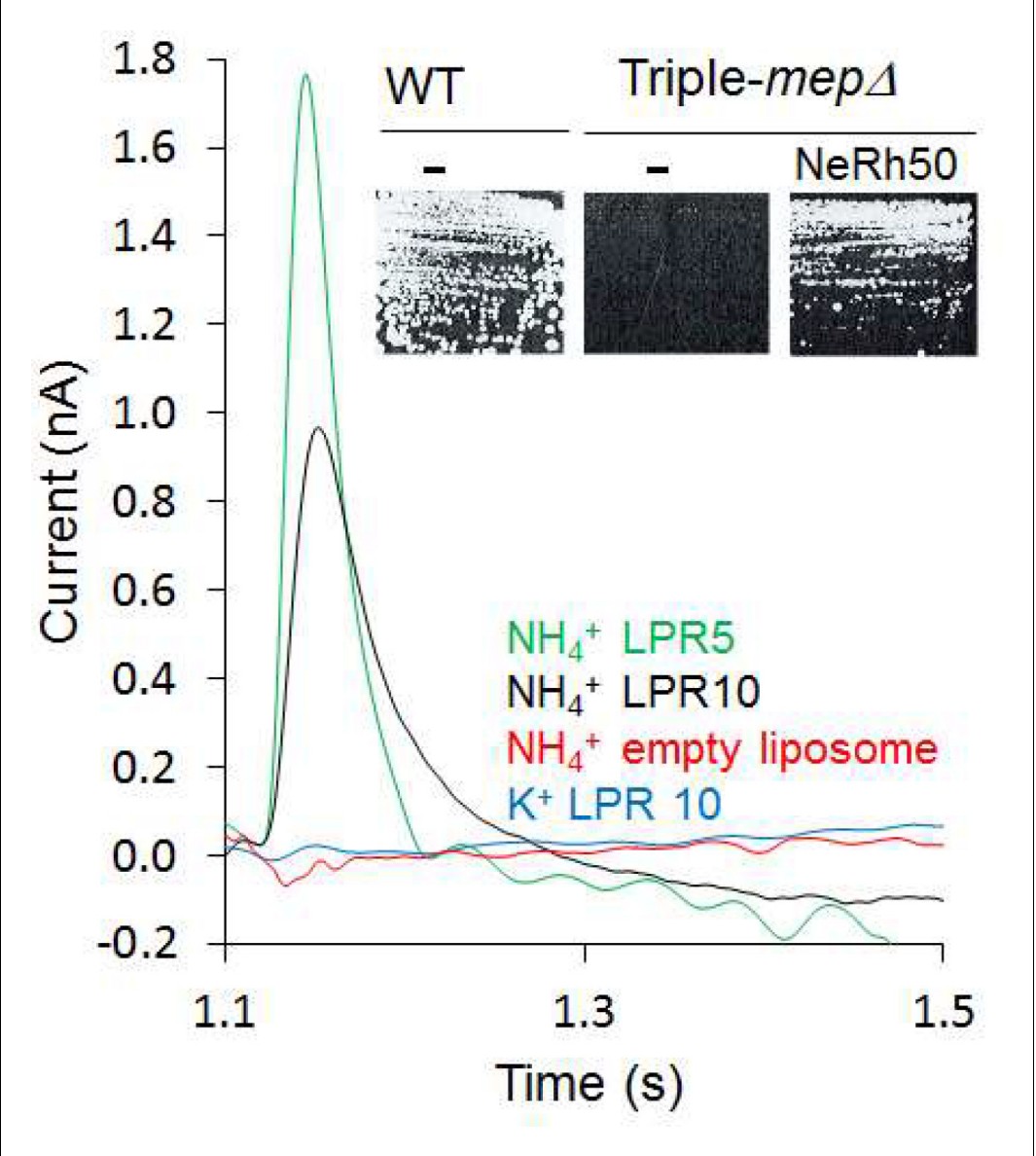

**Figure 1.** Characterization of the activity of NeRh50. Transient current measured using SSME after a 200 mM pulse (ammonium or potassium). *Insert:* Yeast complementation by NeRh50 (strain 31019b, *mep1Δ mep2Δ mep3Δ ura3*) on minimal medium supplemented with 3 mM ammonium as sole nitrogen source.

The online version of this article includes the following source data and figure supplement(s) for figure 1:

**Source data 1.** Characterization of the activity of NeRh50.

**Figure supplement 1.** Characterization of the activity of NeRh50.

CWW; *Figure 2A*), confirming previous observations (*Lamoureux et al., 2007*). Notably, a new observation we made over longer simulation timescales is the presence of a previously unidentified second water-filled channel (periplasmic water wire - PWW). The PWW spans from residue D160 near the S1 region to the central twin-His motif (*Figure 2A*) and is formed both in simulations without and with applied membrane voltage $V_m$ (*Figure 2A* - *Figure 2—figure supplement 2*; $V_m$ in *E. coli* ~ −140 mV [*Felle et al., 1980*]).

As the protonation pattern of the twin-His motif has been found to play a role in the hydration of the protein (*Ishikita and Knapp, 2007*), two different tautomeric states of the twin-His motif were systematically probed in the simulations. The tautomeric state in which H168 is protonated on its $N_\delta$ and H318 is protonated on its $N_\epsilon$ atom is referred to as 'DE', while 'ED' terms the twin-His

**Table 1.** Decay time constants ($s^{-1}$) of transient currents triggered after an ammonium or potassium pulse of 200 mM in proteoliposomes containing AmtB at various LPR*.

| | $NH_4^+$ | | $K^+$ | |
|---|---|---|---|---|
| **Variant** | **LPR 10** | **LPR 5** | **LPR 10** | **LPR 5** |
| AmtB-WT | 13.4 ± 1.5 | 18.7 ± 1.0 | NC | NC |
| D160A | 21.6 ± 1.2 | 24.3 ± 1.5 | NC | NC |
| D160E | 17.03 ± 2.84 | 19.53 ± 1.8 | NC | NC |
| H168A H318A | 29.5 ± 2.1 | 29.8 ± 2.6 | NC | NC |
| S219A H168A H318A | NC | NC | NC | NC |
| H168A | 28.3 ± 1.5 | 38.0 ± 1.0 | 2.7 ± 0.5 | 5.2 ± 1.0 |
| H318A | 22.56 ± 2.63 | 28.25 ± 3.1 | 10.07 ± 1.7 | 15.64 ± 2.1 |
| NeRh50 | 24.0 ± 1.7 | 39.0 ± 3.6 | NC | NC |

*NC: No transient current recorded.

The online version of this article includes the following source data for Table 1:

**Source data 1.** Decay time constants ($s^{-1}$) of transient currents triggered after an ammonium or potassium pulse of 200 mM measured by SSME.

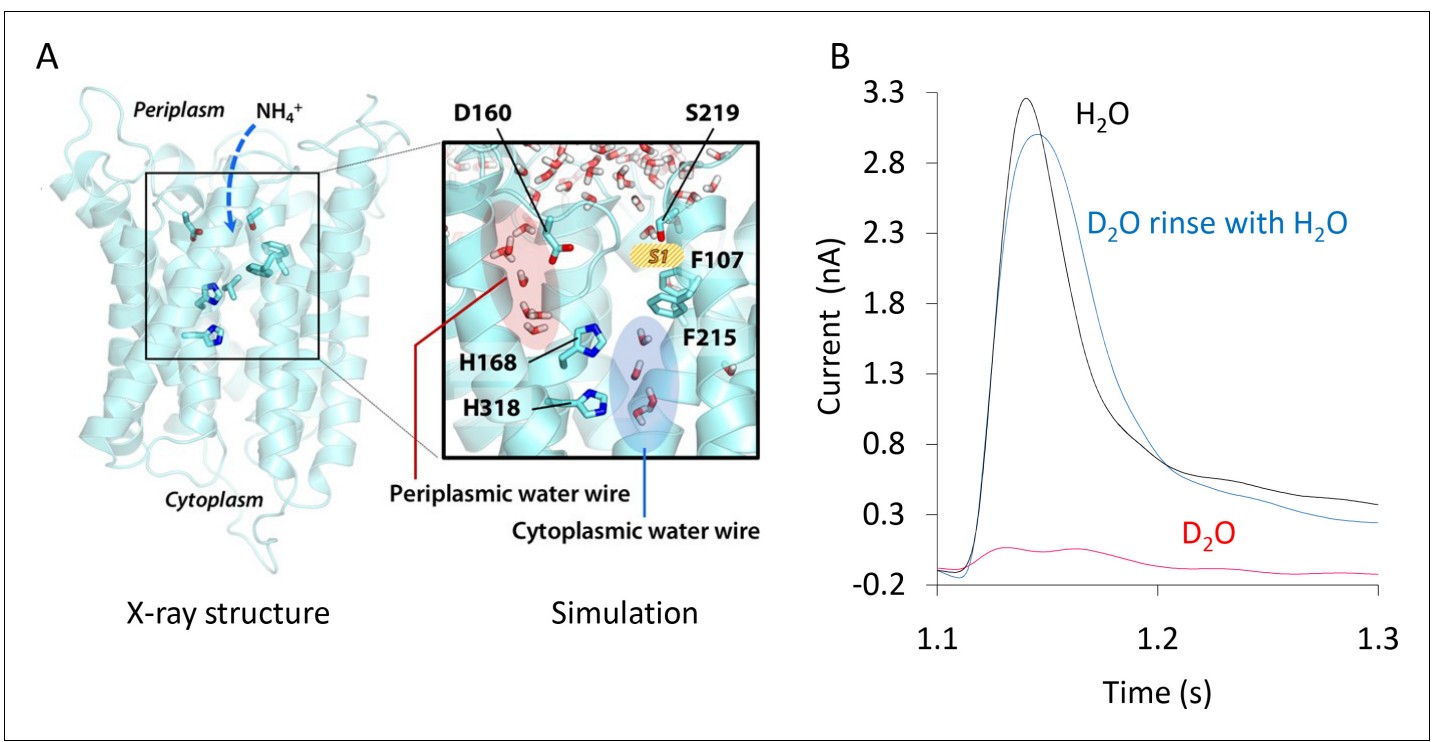

**Figure 2.** Formation and functionality of the periplasmic (PWW) and cytoplasmic (CWW) water wires in AmtB. (**A**) Extended atomistic simulations show a hydration pattern across the protein, in which cytoplasmic and periplasmic water wires, connected via H168, form a continuous pathway for proton transfer from the S1 $NH_4^+$ sequestration region to the cytoplasm. (**B**) Transient currents measured following a 200 mM ammonium pulse on sensors prepared with solutions containing either $H_2O$ (black) or $D_2O$ (red). $D_2O$ sensors were rinsed with $H_2O$ solutions and subsequently exposed to another 200 mM ammonium pulse (blue).

The online version of this article includes the following source data and figure supplement(s) for figure 2:

**Source data 1.** Functionality of the periplasmic (PWW) and cytoplasmic (CWW) water wires in AmtB.
**Figure supplement 1.** Evolutionary conservation of the proton and hydrophobic pathways for $H^+$ and $NH_3$ translocation in AmtB.
**Figure supplement 2.** Evolution and occupancy of the Periplasmic Water Wire (PWW).
**Figure supplement 3.** Evolution and occupancy of the Cytoplasmic Water Wire (CWW).
**Figure supplement 4.** The DE and ED twin-His motif configurations.

configuration where H168 is protonated on $N_\epsilon$ and H318 is protonated on $N_\delta$ (*Figure 2A* - *Figure 2—figure supplement 2*, *3*, *4*). Formation of the CWW is observed to occur within a few nanoseconds at the beginning of each simulation. In the DE tautomeric state, the cytoplasmic pocket of each subunit almost continuously remains occupied by 3–4 water molecules for the rest of the simulation (*Figure 2A* - *Figure 2—figure supplement 2*, *3*, *4*; data for 0 mV membrane voltage). In the ED state, greater fluctuations in the number of water molecules in the chain are seen, and the average occupancy is decreased. Using a cut-off value of three water molecules per subunit, a complete water chain is present during 79% of the simulations in the DE state, and only during 12% of the simulated time in the ED state. The PWW is generally more transiently occupied than the cytoplasmic channel; however, we record up to 23% occupancy with at least three water molecules when the histidine sidechains are in the ED tautomeric state (*Figure 2A* - *Figure 2—figure supplement 2*, *3*, *4*).

Both water wires are connected via the twin-His motif, which bridges the aqueous chains, while preventing the formation of a continuous water channel in the simulations. Although neither the CWW nor the PWW are sufficiently wide to allow the transfer of solvated $NH_4^+$, water molecules and histidine side chains could serve as efficient pathways to facilitate proton transfer in proteins (*Acharya et al., 2010*). As shown in *Figure 2—figure supplement 1*, the key residues that line both water wires in AmtB are highly conserved in the family.

## The interconnected water wires are functionally essential to AmtB activity

To experimentally test if the water wires are essential for proton conduction during the AmtB transport cycle, we made use of the reduced deuteron mobility of heavy water $D_2O$. Because deuterons have twice the mass of a proton and the bond strength is increased, the deuteron mobility is reduced by 30% for each $D_2O$ molecule compared to normal water (*Wiechert and Beitz, 2017*). Since the polar network of water we identified involves more than three water molecules (*Figure 2A*), AmtB should be nearly inactive if tested in the presence of $D_2O$. Indeed, we found that in an SSME-based assay where all buffers used to prepare the proteoliposomes and SSM sensors were made using $D_2O$, AmtB activity was completely abolished compared to buffer containing water (*Figure 2B*). After rinsing the sensor prepared in $D_2O$ with water, AmtB re-gained 100% of its activity measured by SSME, showing that the presence of $D_2O$ did not affect the protein itself or the integrity of the proteoliposomes (*Figure 2B*). Further calculations suggested that $H^+$ transfer between the water molecules is possible both within the PWW and CWW and could occur with high rates (the highest energy barrier is ~18 kJ/mol in the cytoplasmic wire near the twin-His motif; *Table 2*). Taken together, the experimental and computational data suggest that proton transfer between water molecules, most likely the PWW and CWW detected in the simulations, may underpin the electrogenic activity of AmtB.

## AmtB activity is not driven by the proton motive force

In the absence of ammonium, a proton pulse did not trigger a discernible current and additionally, in the presence of ammonium, an inward-orientated pH gradient did not increase AmtB activity (*Figure 3*). These findings suggest that there is no $H^+$-dependent symport activity of AmtB, showing instead that AmtB is not able to translocate a proton in the absence of $NH_4^+$, and indicating that the current induced by AmtB activity is generated by specific deprotonation of the substrate and subsequent $H^+$ translocation. Furthermore, they show that AmtB cannot act as an uncoupler, which raises the question of proton selectivity and the coupling between $NH_3$ and $H^+$ transfer (*Boogerd et al., 2011*; *Maeda et al., 2019*). Our current data suggest that the PWW is transiently occupied and that its occupancy is strongly dependent on the particular state and conformation of D160, since even a D to E conservative change abolished presence of the PWW (*Figure 4A*). Any disruption of the PWW will, in turn, impede the capability of AmtB to transfer $H^+$. The functionally relevant conformation and protonation state of D160 that stabilize the PWW is likely to be coupled to the presence of a charged substrate binding near S1, thereby linking substrate binding and deprotonation to $H^+$ transfer.

**Table 2.** Free energies for proton translocation through the cytoplasmic and periplasmic water wires and neighboring water molecules (bulk)*.

| | | | | Z (Å) | Free energy (kJ/mol) |
|---|---|---|---|---|---|
| (bulk) | Peripl. water wire | | wat1 | 14.7 | 0.0 |
| | | | wat2 | 12.7 | 8.7 |
| | | | wat3 | 10.7 | 15.0 |
| | | | wat4 | 8.3 | 14.4 |
| | | | wat5 | 6.1 | 7.5 |
| D160 | | | wat6 | 5.4 | 11.0 |
| | | | wat7 | 3.2 | 14.4 |
| | | | wat8 | 0.6 | 18.5 |
| H168 | | | | | |
| | cytopl. water wire | | wat9 | −0.4 | 17.3 |
| | | | wat10 | −0.8 | 14.4 |
| | | | wat11 | −3.2 | 12.1 |
| H318 | | | wat12 | −5.1 | 13.8 |

*The vertical coordinate z was calculated relative to the position of the sidechain of H168. Positions of the sidechains of D160, H168 and H318 with respect to the periplasmic and cytoplasmic water wires are indicated in the left column.

## The residue D160 is essential to stabilize the PWW

As the PWW is formed near the sidechain of D160, an invariant residue in the Amt/Mep/Rh super-family (*Marini et al., 2006*; *Thomas et al., 2000*), we further investigated the role of this residue in ensuring PWW and CWW stability by simulating the AmtB D160A and D160E mutants. Both mutants were stable on the time scale of our simulations and we did not detect major rearrangements in the protein. Moreover, all the elution profiles of the purified WT and variants proteins obtained by analytical size exclusion chromatography, before and after solubilization of the proteoliposomes in 2% DDM, were identical, showing a single monodisperse peak eluting between 10.4–10.6 ml (*Figure 4—figure supplement 1*). Taken together, these results suggest that major structural re-arrangements in the mutants are unlikely to occur. The simulations revealed no difference in the formation of the CWW in the D160A and D160E variants compared to the WT, however the formation of the PWW is almost completely abolished in the presence of these mutations (*Figure 4A*).

We then expressed wild-type AmtB as well as the D160A and D160E mutants in *S. cerevisiae* triple-*mepΔ*. Using ammonium as the sole nitrogen source, we found that cells expressing the mutants failed to grow, showing that AmtB[D160A] or AmtB[D160E] are unable to replace the function of the endogenous Mep transporters (*Figure 4B*).

The activity of the purified variants reconstituted into liposomes was next quantified using SSME. Electrogenic transport activity, triggered by a 200 mM ammonium pulse, led to a transient current with a maximum amplitude of 3.38 nA in AmtB, while AmtB[D160A] and AmtB[D160E] displayed reduced maximum currents of 0.63 nA and 1.42 nA respectively (*Figure 4B*, *Figure 4—figure supplement 2*). Importantly, the lifetime of currents in both variants was unaffected by changes in liposomal LPR, and therefore the small transient current accounts for the binding of a $NH_4^+$ to the proteins and not a full translocation cycle (*Table 1*, *Figure 4—figure supplement 2*; *Bazzone et al., 2017*). Additionally, it was impossible to determine with confidence a catalytic constant ($K_m$) for both variants since no clear saturation was reached, even after an ammonium pulse of 200 mM (*Figure 4C*). These results thus demonstrate that AmtB[D160A] and AmtB[D160E] are transport-deficient. Our data show that residue D160 plays a central role in the transport mechanism as opposed to having a strictly structural role as previously suggested (*Khademi et al., 2004*). Moreover, the fact that the conservative D to E variation at position 160 impairs ammonium transport via AmtB indicates that D160 does not only show electrostatic interaction with $NH_4^+$ at the S1 site but is also involved in the translocation mechanism by stabilizing the PWW.

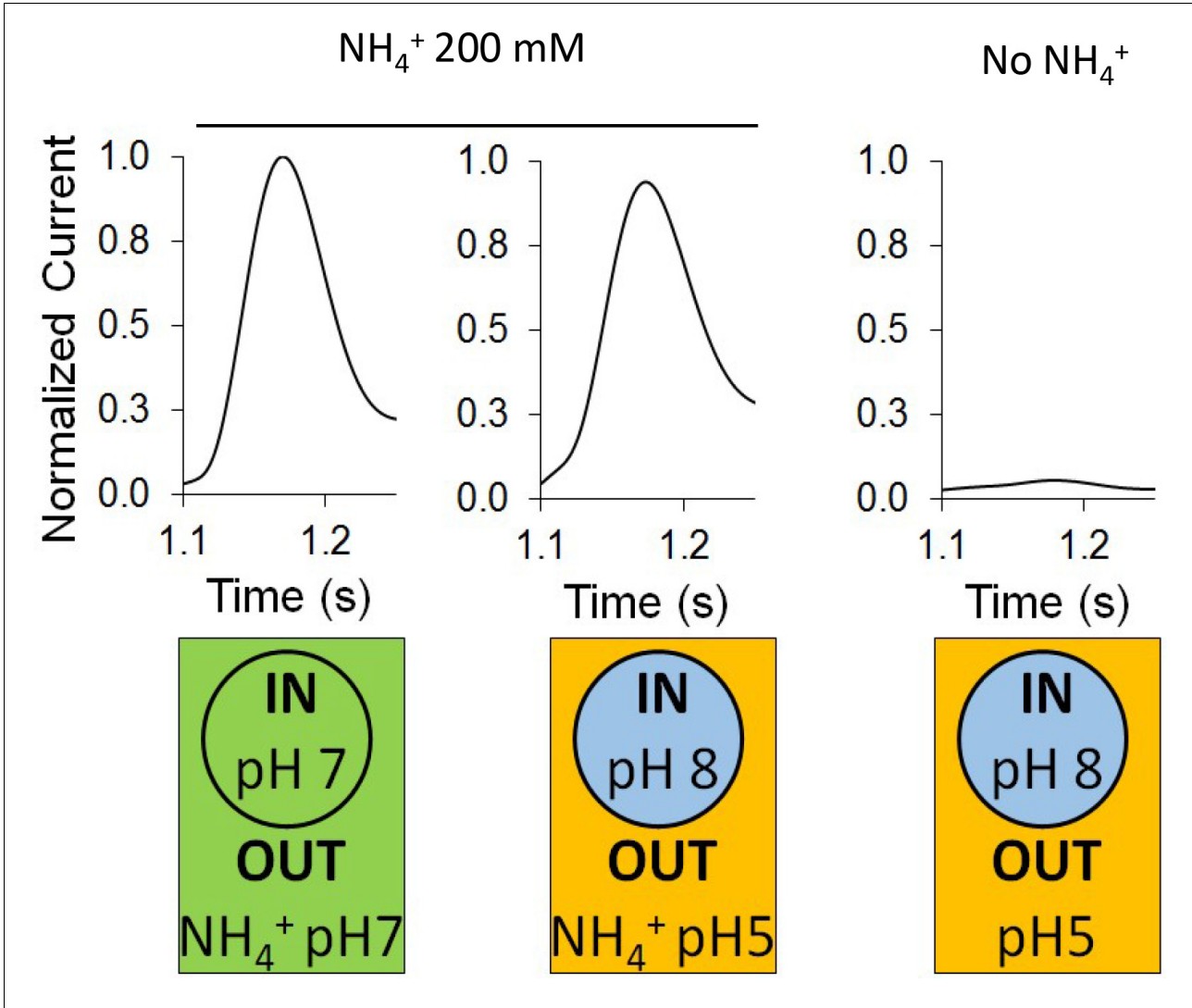

**Figure 3.** Effect of a proton gradient on AmtB activity. The transient currents were measured using SSME following an ammonium pulse of 200 mM at pH 7 (left) or under an inwardly directed pH gradient in the presence (center) or absence (right) of ammonium. eight sensors from two independent protein purification batches were measured, with three measurements recorded for each sensor. Single representative traces were chosen to visualize the results. Each sensor was measured in the order pH (in/out) 7/7, 8/5, 8/5 (this time without $NH_4^+$), and finally 7/7 again to be sure that the signals do not significantly decrease with time. The data are normalized against the measurements done at pH7 in/out for each sensor.

The online version of this article includes the following source data for figure 3:

**Source data 1.** Effect of a proton gradient on AmtB activity.

## AmtB switches from transporter to channel-like activity in the absence of the 'twin-His' motif

The CWW and PWW are connected via the twin-His motif, which bridges the aqueous chains, while preventing the formation of a continuous water channel in the simulations (*Figure 2A*). We therefore next probed if the twin-His motif enables proton transfer between the two water wires by recording the activities of twin-His variants. Expressed in *S. cerevisiae* triple-*mepΔ*, AmtB[H168A/H318A] did not support cell growth on low ammonium (*Figure 5A*). *In-vitro* SSME measurements with this variant displayed LPR-independent current decay rates (*Figure 5A*, *Figure 5—figure supplement 1*, *Table 1*), showing that the residual current is caused by the association of $NH_4^+$ to AmtB without further transport. No current was recorded for the triple mutant AmtB[S219A/H168A/H318A], in which binding at the periplasmic face was further altered, confirming that the residual current reflects $NH_4^+$ interaction near S1 (*Figure 5—figure supplement 1*). The double-His mutant AmtB[H168A/H318A] is

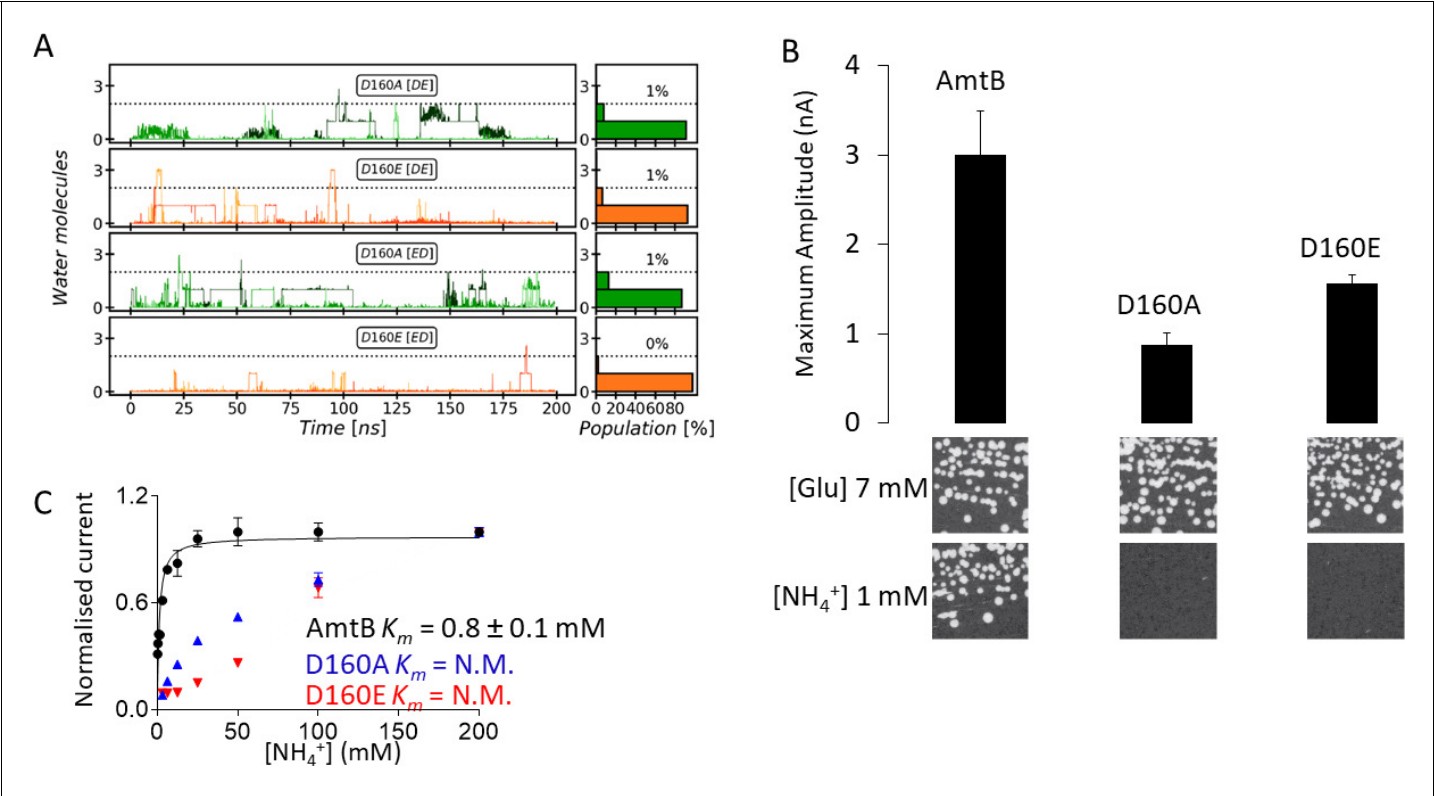

**Figure 4.** Effect of D160 substitutions. **(A)** The Periplasmic Water Wire (PWW) in the D160A and D160E variants. We observe no significant occupancy of the PWW above the threshold of at least three water molecules in the D160A and D160E AmtB variants, irrespective of the tautomeric protonation states of H168 and H318 (DE or ED, see Materials and method section). **(B)** *Upper panel:* maximum amplitude of the transient current measured using SSME following a 200 mM ammonium pulse. Eight sensors from two independent protein purification batches were measured, with three measurements recorded for each sensor (means ± SD). *Lower panel*: yeast complementation test (strain 31019b, *mep1Δ mep2Δ mep3Δ ura3*) using 7 mM Glutamate (Glu) or 1 mM ammonium as a sole nitrogen source. The growth tests have been repeated twice. **(C)** Kinetics analysis of the transport of ammonium. The maximum amplitudes recorded after a 200 mM ammonium pulse have been normalized to 1.0 for comparison. N.M.: Non Measurable. eight sensors from two independent protein purification batches were measured, with three measurements recorded for each sensor (means ± SD). The online version of this article includes the following source data and figure supplement(s) for figure 4:

**Source data 1.** Effect of D160 substitutions on AmtB activity measured by SSME.
**Figure supplement 1.** Size Exclusion Chromatography analysis of AmtB.
**Figure supplement 2.** Characterization of the activity and specificity of AmtB variants.

thus able to interact with $NH_4^+$ but cannot transport the substrate across the membrane. This supports our previous structural analysis showing that the CWW in the pore of the double-His mutant AmtB[H168A/H318A] is absent (*Javelle et al., 2006*).

By contrast, the two single histidine-to-alanine substitutions in the twin-His motif unexpectedly produced an LPR-dependent current in our SSME recordings (*Figure 5A*, *Figure 5—figure supplement 1*, *Table 1*). Furthermore, triple-*mepΔ* yeast cells expressing these variants were able to grow in the presence of low ammonium concentrations (*Figure 5A*). Our previous crystal structure (*Javelle et al., 2006*) and our MD simulations (*Figure 5—figure supplement 2*) show increased hydration in the area around A168, which could potentially form a pathway for direct translocation of $NH_4^+$ without a deprotonation step. To test this hypothesis, we measured the activity of AmtB[H168A] and AmtB[H318A] in $D_2O$ conditions, as described above. Crucially, the activity of both variants measured in the presence or absence of $D_2O$ was similar (*Figure 5B*), in contrast to native AmtB where no activity was recorded in $D_2O$ (*Figure 2B*), showing that proton transfer between water molecules is not a key mechanistic feature in the activity of the mutants. Additionally, the translocation of $NH_4^+$ is not saturable in the tested concentration range [12.5–200 mM] for AmtB[H168A] and AmtB[H318A] (*Figure 5C*). Summarizing, these results suggest that AmtB switches from

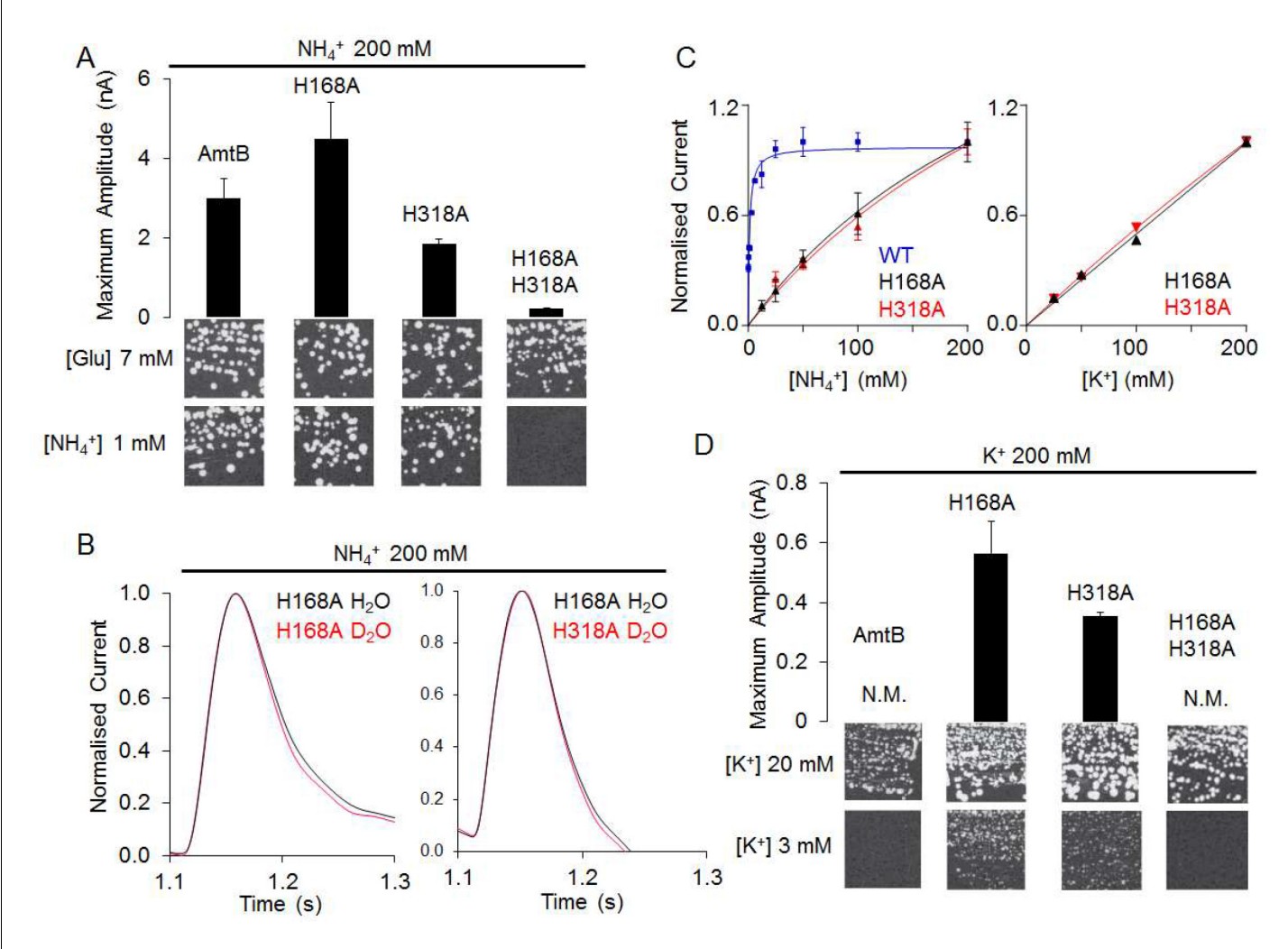

**Figure 5.** The AmtB[H168A] and AmtB[H318A] lose their specificity toward ammonium. (**A**) *Upper panels:* maximum amplitude of the transient current measure using SSME after a 200 mM ammonium pulse. Eight sensors from two independent protein purification batches were measured, with three measurements recorded for each sensor (means ± SD). *Lower panels:* yeast complementation test (strain 31019b, *mep1Δ mep2Δ mep3Δ ura3*) using 7 mM Glutamate (Glu) or 1 mM ammonium as a sole nitrogen source. The growth tests have been repeated twice. (**B**) Transient currents measured using SSME following a 200 mM ammonium pulse on sensors prepared with solutions containing either $H_2O$ (black) or $D_2O$ (red). The maximum amplitudes recorded after a 200 mM ammonium pulse on sensor prepared in $H_2O$ have been normalized to 1.0 for comparison. eight sensors from two independent protein purification batches were measured, with three measurements recorded for each sensor (means ± SD). (**C**) Kinetics analysis of the transport of $NH_4^+$ (or $K^+$ in AmtB[H168A] (black), AmtB[H318A] (red) and WT-AmtB (bleu, only for $NH_4^+$, as no signal was measurable with $K^+$). The maximum amplitudes recorded after a 200 mM $NH_4^+$ or $K^+$ pulse have been normalized to 1.0 for comparison. Eight sensors from two independent protein purification batches were measured, with three measurements recorded for each sensor (means ± SD). (**D**) *Upper panels:* maximum amplitude of the transient current measured using SSME after a 200 mM potassium pulse. N.M. Non Measurable. Eight sensors from two independent protein purification batches were measured, with three measurements recorded for each sensor (means ± SD). *Lower panels:* yeast complementation test (strain #228, *mep1Δ mep2Δ mep3Δ trk1Δ trk2Δ leu2 ura3*) using media supplemented with 20 mM or 3 mM KCl. The growth test has been repeated twice.

The online version of this article includes the following source data and figure supplement(s) for figure 5:

**Source data 1.** Effect of H168 and/or H318 substitution on AmtB activity and selectivity measured by SSME.

**Figure supplement 1.** Characterization of the activity and specificity of AmtB variants.

**Figure supplement 2.** MD simulation of AmtB[H168A] showing formation of a continuous water wire traversing the central pore region.

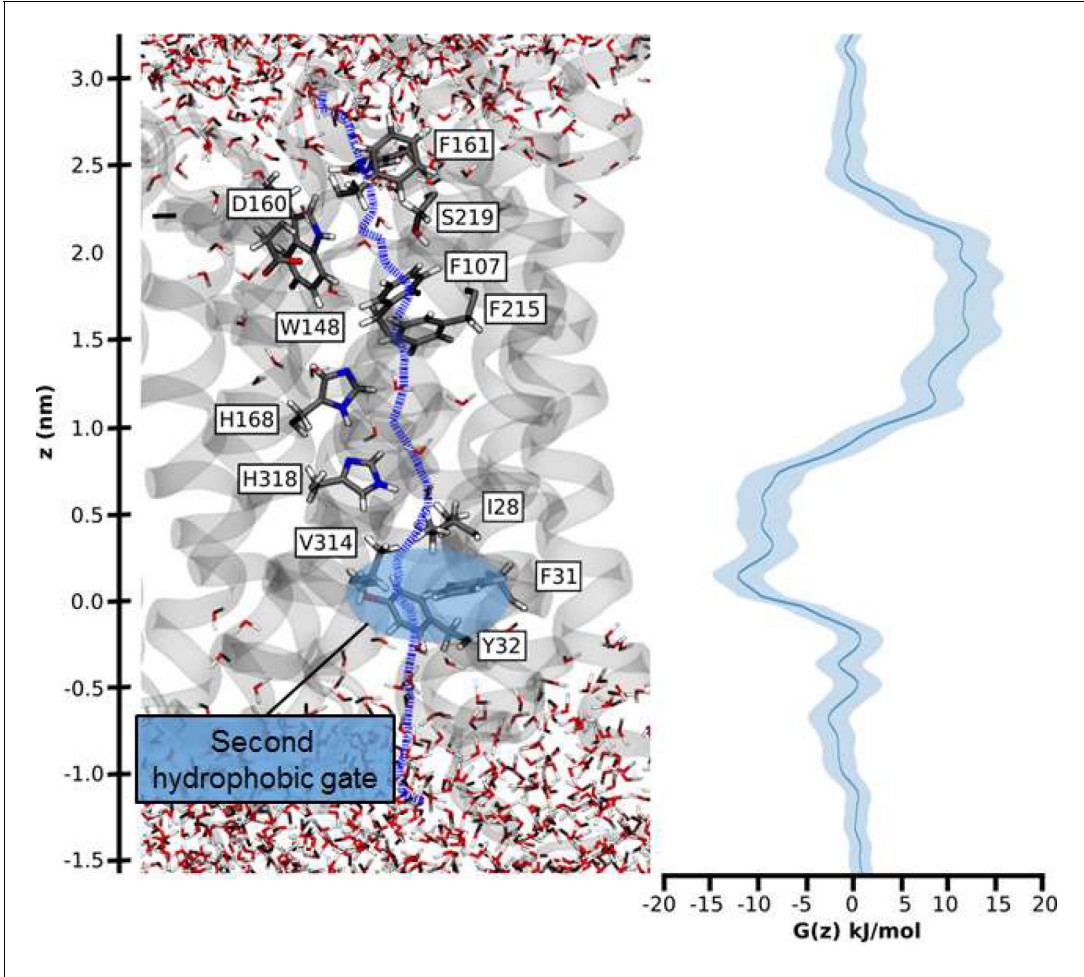

**Figure 6.** Hydrophobic pathway and energetics for NH$_3$ translocation in AmtB. We probed an optimal pathway for NH$_3$ transfer during our PMF calculations (left, purple dash trajectory) in the presence of both the PWW and CWW. The software HOLE (49) was used to determine the most likely transfer route. The pathway from the periplasm to the cytoplasm traverses the hydrophobic gate region (F107 and F215), crosses the cavity next to the twin-His motif (H168 and H318) occupied by the CWW, and continues across a second hydrophobic region (I28, V314, F31, Y32) before entering the cytoplasm.

transporter- to channel-like activity in the absence of the twin-His motif, directly translocating hydrated NH$_4^+$ through the pore. In the wild-types of Amt/Mep/Rh protein family members, the twin-His motif is highly conserved, which shows that transporter, as opposed to channel activity, is mechanistically crucial for the function of these proteins. The only variation seen in naturally occurring sequences is a replacement of the first His by Glu in some fungal Mep proteins (*Javelle et al., 2006*; *Thomas et al., 2000*). Channel activity is so far only observed for the alanine mutants, not the wild-type. We hypothesized that transport activity might thus be key to ensure ion selectivity of AmtB, since NH$_4^+$ and K$^+$ are cations of similar size and hydration energy (*Aydin et al., 2020*).

## The twin-His motif interconnects the two water wires to ensure the selectivity of AmtB

Since NH$_4^+$ was directly translocated in the absence of the twin-his motif and earlier studies implicated a role of the motif in AmtB selectivity (*Ganz et al., 2020*; *Hall and Yan, 2013*), we repeated our SSME experiments on the AmtB$^{H168A}$ and AmtB$^{H318A}$ variants using the competing K$^+$ ion as substrate. A 200 mM K$^+$ pulse triggered currents in both variants, whose decay rates strongly depended on the LPR (*Figure 5D*, *Table 1*, *Figure 5—figure supplement 1*). Furthermore, the single His variants, but not native AmtB, complemented the growth defect of a yeast strain lacking its three endogenous ammonium (Mep) and potassium (Trk) transporters when a limited concentration

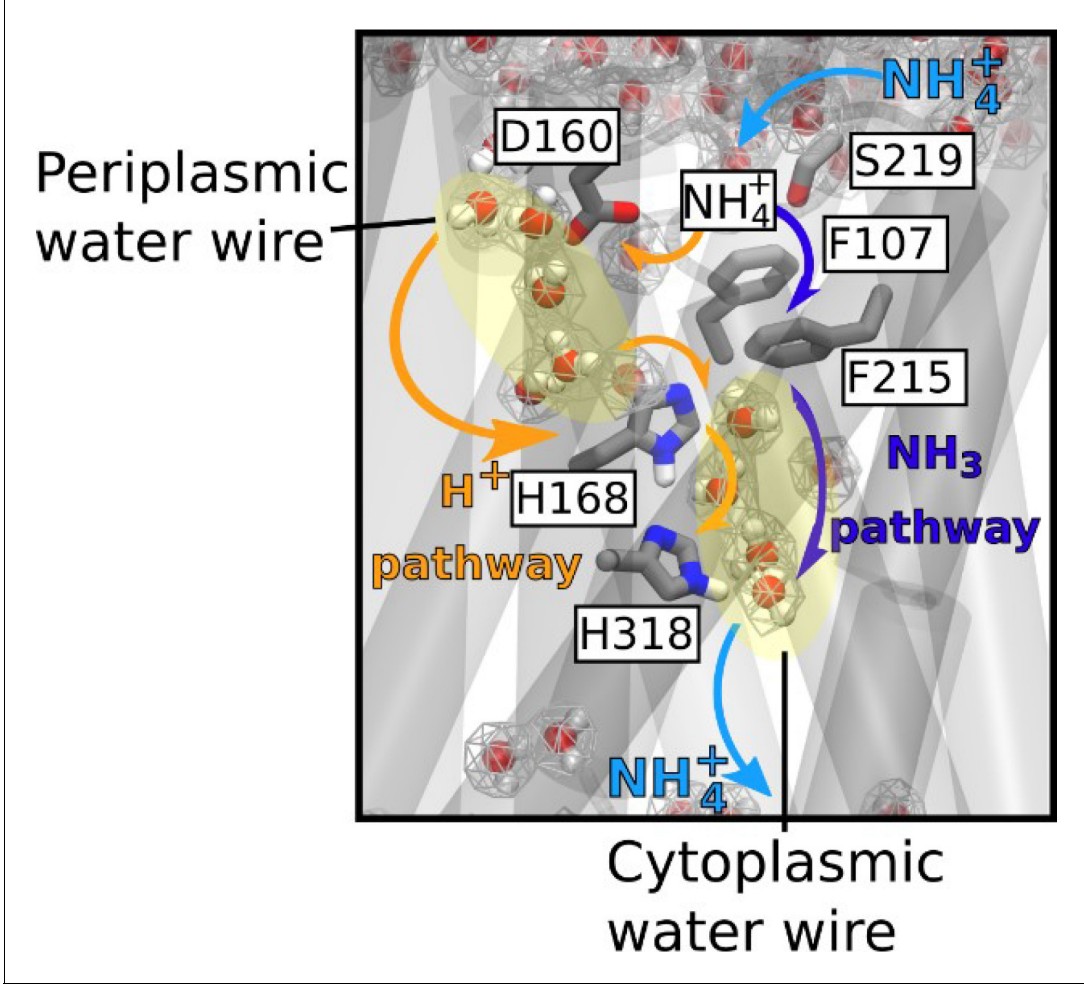

**Figure 7.** Mechanism of electrogenic $NH_4^+$ translocation in AmtB. Following sequestration of $NH_4^+$ at the periplasmic face, $NH_4^+$ is deprotonated and $H^+$ and $NH_3$ follow two separated pathways to the cytoplasm (orange arrows depict the pathway for $H^+$ transfer, dark blue arrows for $NH_3$), facilitated by the presence of two internal water wires. $NH_3$ reprotonation likely occurs near the cytoplasmic exit (*Figure 6*). The hydrated regions within the protein as observed in simulations are highlighted by wireframe representation, crucial residues involved in the transport mechanism are shown as sticks.

of $K^+$ was present (*Figure 5D*). The $K^+$ translocation activity is not saturable in the substrate range [12.5–200 mM] (*Figure 5C*). These results demonstrate that both variants, AmtB[H168A] and AmtB[H318A], translocate $K^+$ ions across the membrane. The substitutions within the twin-His motif thus abolished selectivity for $NH_4^+$.

The presence of both histidine residues is therefore critical in ammonium transport, since permeability of ammonium transporters for $K^+$ would compromise ionic homeostasis and disrupt the membrane potential of *E. coli* cells, which crucially depends on maintaining $K^+$ concentration gradients across the membrane. Moreover, since AmtB is expressed in *E. coli* under nitrogen starvation conditions (low $NH_4^+/K^+$ ratio), loss of selectivity for $NH_4^+$ would impede ammonium uptake. Our results thus demonstrate that the twin-His motif, which is highly conserved amongst members of the family, is an essential functional element in the transport mechanism, preventing the transport of competing cations, whilst providing a pathway for proton transfer by bridging the periplasmic and the cytoplasmic water wires.

### $NH_3$ permeation through the hydrophobic pore

Umbrella sampling free-energy calculations were performed to establish the rate limiting step of $NH_4^+$ transport. Our calculations show that $NH_3$ translocation experiences only a moderate energy barrier (~10 kJ/mol) at the periplasmic hydrophobic constriction region (F107 and F215) (*Figure 6*).

The starting points of the sampling windows were determined from the centers of HOLE calculations (*Smart et al., 1996*), optimizing the pathway of $NH_3$ translocation across the pore. A possible influence from this selection regarding the pathway was further reduced by allowing the molecule to move freely perpendicular to the pore axis within a radius of 5 Å in addition to extensive sampling; however, residual bias from window selection cannot be completely excluded. From the free energy profile of $NH_3$ translocation, we identified a shallow binding site below the twin-His motif (~5 kJ/mol). This is followed by a second hydrophobic region (I28, V314, F31 and Y32) that forms a small energy barrier between this binding site and the cytoplasmic exit. The increased residence time of $NH_3$ within this energy minimum suggests that reprotonation to $NH_4^+$, caused by the cytoplasmic pH, occurs in this region (*Figure 6*). Since both energy barriers for $H^+$ transfer along the water chains and $NH_3$ permeation are relatively small, we concluded that either initial deprotonation or proton transfer across the twin-His motif could be rate-limiting for overall $NH_4^+$ transport.

## Conclusion

A new model for the mechanism of electrogenic ammonium transport therefore emerges from our findings (*Figure 7*). After deprotonation of $NH_4^+$ at the periplasmic side, a previously undiscovered polar conduction route enables $H^+$ transfer into the cytoplasm. A parallel pathway, lined by hydrophobic groups within the protein core, facilitates the simultaneous transfer of uncharged $NH_3$, driven by concentration differences. On the cytoplasmic face, the pH of the cell interior leads to recombination to $NH_4^+$, most likely near a second hydrophobic gate (*Figure 6*). The twin-His motif, which bridges the water chains constitutes the major selectivity gate for $NH_4^+$ transport preventing $K^+$ flow. We propose that this mechanism is conserved amongst the electrogenic members of the Amt/Mep/Rh family. Importantly, two RhAG polymorphisms associated to the overhydrated stomatocytosis human syndrome have also acquired the ability to transport $K^+$. Thus, deciphering the transport mechanism of two archetypal members of the family such as AmtB and NeRh50 could bring new insights to the understanding of substrate specificity determinants in Rh proteins in the context of human diseases (*Bruce et al., 2009*).

Our findings define a new mechanism, by which ionizable molecules that are usually charged in solution are selectively and efficiently transported across a highly hydrophobic environment like the AmtB/Rh pore. Alongside size-exclusion and ion desolvation (*Kopec et al., 2018*), it adds a new principle by which selectivity against competing ions can be achieved. Other biological transport systems, like the formate/nitrite transporters, may share similar mechanisms involving deprotonation-reprotonation cycles (*Wiechert and Beitz, 2017*).

## Materials and methods

### Key resources table

| Reagent type (species) or resource | Designation | Source or reference | Identifiers | Additional information |
|---|---|---|---|---|
| Gene (*Escherichia coli*) | AmtB | *Zheng et al., 2004* | Uniprot: C3TLL2 | |
| Gene (*Nitrosomonas europea*) | Rh50 | *Lupo et al., 2007* | Uniprot Q82 × 47 | |
| Strain, strain background (*Escherichia coli*) | C43 (DE3) | *Miroux and Walker, 1996* | | Chemically competent cells |
| Strain, strain background (*Escherichia coli*) | GT1000 | *Javelle et al., 2004* | | Chemically competent cells |
| Recombinant DNA reagent | pET22b (+) | Novagen | Cat# - 69744 | |
| Recombinant DNA reagent | pDR195 | *Rentsch et al., 1995* | Addgene - 36028 | High copy yeast expression vector |
| Recombinant DNA reagent | pAD7 | *Cherif-Zahar et al., 2007* | | pESV2-RH50(His)$_6$ |

*Continued on next page*

*Continued*

| Reagent type (species) or resource | Designation | Source or reference | Identifiers | Additional information |
|---|---|---|---|---|
| Recombinant DNA reagent | p426MET25 | *Mumberg et al., 1994* | | |
| Recombinant DNA reagent | PZheng | *Zheng et al., 2004* | | pET22b-AmtB(His)$_6$ |
| Recombinant DNA reagent | pGDM2 | This study | | pET22b-AmtB(His)$_6^{H168AH318A}$ |
| Recombinant DNA reagent | pGDM4 | This study | | pET22b-AmtB(His)$_6^{D160A}$ |
| Recombinant DNA reagent | pGDM5 | This study | | pET22b-AmtB(His)$_6^{D160E}$ |
| Recombinant DNA reagent | pGDM6 | This study | | pET22b-AmtB(His)$_6^{S219AH168AH318A}$ |
| Recombinant DNA reagent | pGW2 | This study | | pET22b-AmtB(His)$_6^{H168A}$ |
| Recombinant DNA reagent | pGDM9 | This study | | pDR195-AmtB(His)$_6^{D160A}$ |
| Recombinant DNA reagent | pGDM10 | This study | | pDR195-AmtB(His)$_6^{D160E}$ |
| Recombinant DNA reagent | pGDM12 | This study | | pDR195-AmtB(His)$_6^{H168AH318A}$ |
| Recombinant DNA reagent | pGDM13 | This study | | pDR195-AmtB(His)$_6^{S219AH168AH318A}$ |
| Recombinant DNA reagent | pGW7 | This study | | pDR195-AmtB(His)$_6^{H168A}$ |
| Sequence-based reagent | AmtB$^{S219A}$ F | IDT | PCR Primer (Mutagenesis) | GGTGGCACCGTGGTGG**ATA**TTAACGCCGCAATC |
| Sequence-based reagent | AmtB$^{D160A}$ F | IDT | PCR Primer (Mutagenesis) | CTCACGGTGCGCTGG**CC**TTCGCGGGTGGCACC |
| Sequence-based reagent | AmtB$^{D160E}$ F | IDT | PCR Primer (Mutagenesis) | CTCACGGTGCGCTGG**AG**TTCGCGGGTGGCACC |
| Sequence-based reagent | AmtB$^{H168A}$ F | IDT | PCR Primer (Mutagenesis) | GGTGGCACCGTGGTGG**CCA**TTAACGCCGCAATC |
| Sequence-based reagent | AmtB$^{H318A}$ F | IDT | PCR Primer (Mutagenesis) | TGTCTTCGGTGT**GGC**CGGCGTTTGTGGCATT |
| Sequence-based reagent | AmtB XhoI | IDT | PCR primer | AGTC**CTCGAG**ATGAAGATAGCGACGATAAAA |
| Sequence-based reagent | AmtB BamHI | IDT | PCR primer | AGTC**GGATCC**TCACGCGTTATAGGCATTCTC |
| Sequence-based reagent | P5'NeRh | IDT | PCR primer | GCC**ACTAGT**ATGAGTAAACACCTATGTTTC |
| Sequence-based reagent | P3'NeRh | IDT | PCR primer | GCC**GAATTC**CTATCCTTCTGACTTGGCAC |
| Peptide, recombinant protein | AmtB(His)$_6$ | This study | | purified from *E. coli* C43 (DE3) cells |
| Peptide, recombinant protein | AmtB(His)$_6^{D160A}$ | This study | | purified from *E. coli* C43 (DE3) cells |
| Peptide, recombinant protein | AmtB(His)$_6^{D160E}$ | This study | | purified from *E. coli* C43 (DE3) cells |
| Peptide, recombinant protein | AmtB(His)$_6^{H168AH318A}$ | This study | | purified from *E. coli* C43 (DE3) cells |

*Continued on next page*

*Continued*

| Reagent type (species) or resource | Designation | Source or reference | Identifiers | Additional information |
|---|---|---|---|---|
| Peptide, recombinant protein | AmtB (His)$_6$$^{S219AH168AH318A}$ | This study | | purified from *E. coli* C43 (DE3) cells |
| Peptide, recombinant protein | AmtB(His)$_6$$^{H168A}$ | This study | | purified from *E. coli* C43 (DE3) cells |
| Peptide, recombinant protein | AmtB(His)$_6$$^{H318A}$ | This study | | purified from *E. coli* C43 (DE3) cells |
| Peptide, recombinant protein | *Ne*Rh50(His)$_6$ | This study | | purified from *E. coli* C43 GT1000 cells |
| Peptide, recombinant protein | XhoI | Promega | Cat# - R6161 | |
| Peptide, recombinant protein | BamHI | Promega | Cat# - R6021 | |
| Commercial assay or kit | Quikchange XL site-directed mutagensis kit | Agilent Technologies | Cat# 200516 | |
| Chemical compound, drug | n-dodecyl-β-D-maltopyranoside (DDM) | Avanti | Cat#- 850520 | |
| Chemical compound, drug | lauryldecylamine oxide (LDAO) | Avanti | Cat#- 850545 | |
| Chemical compound, drug | *E. coli* Polar Lipids | Avanti | Cat#−100600 | |
| Chemical compound, drug | Phosphotidylcholine (POPC) | Avanti | Cat#−850457 | |
| Software, algorithm | Graphpad Prism software | GraphPad Prism (https://www.graphpad.com) | | Version 6.01 |
| Software, algorithm | Origin Pro Software | Origin Labs (https://www.originlab.com) | | Origin 2017 Version 94E |
| Software, algorithm | SURFE$^2$R Control Software | Nanion (https://www.nanion.de/en/) | | V1.5.3.2 |

## Mutagenesis

AmtB mutants were generated using the Quikchange XL site-directed mutagenesis kit (Agilent Technologies), according to the manufacturer's instructions. The primers used for mutagenesis are listed in Key resources table. The template was the *amtB* gene cloned into the plasmid pET22b(+), as previously described (*Zheng et al., 2004*; Key resources table).

## AmtB and NeRh50 expression in yeast and complementation test

The different variants of *amtB* were amplified using *amtB* cloned into pET22b(+) (Key resources table) as a template with the primers AmtB XhoI and AmtB BamHI (Key resources table) and then sub-cloned into the plasmids pDR195 (Key resources table). The NeRh50 gene was amplified from *N. europaea* genomic DNA (kind gift from Daniel J. Arp and Norman G. Hommes, Department of Botany and Plant Pathology, Oregon State University, Corvallis, USA) using the primers P5'NeRh and P3'NeRh (Key resources table), and was then cloned into the SpeI and EcoRI restriction sites of the high-copy vector p426Met25 (Key resources table), allowing controlled-expression of NeRh50 by the yeast methionine repressible MET25 promoter.

*Saccharomyces cerevisiae* strains used in this study are the 31019b strain (*mep1Δ mep2Δ mep3Δ ura3*) and the #228 strain (*mep1Δ mep2Δ mep3Δ trk1Δ trk2Δ leu2 ura3*) (*Hoopen et al., 2010*; *Marini et al., 1997*). The plasmids used in this study are listed in Key resources table. Cell transformation was performed as described previously (*Gietz et al., 1992*). For growth tests on limiting ammonium concentrations, yeast cells were grown in minimal buffered (pH 6.1) medium and for growth tests on limiting potassium concentrations, a minimal buffered (pH 6.1) medium deprived of potassium salts was used (*Jacobs et al., 1980*). 3% glucose was used as the carbon source and, 0.1% glutamate, 0.1% glutamine or $(NH_4)_2SO_4$ at the specified concentrations were used as the nitrogen sources.

All growth experiments were repeated at least twice.

## Protein purification

AmtB(His$_6$) cloned into the pET22b(+) vector (Key resources table) was overexpressed and purified as described previously (*Zheng et al., 2004*). The plasmid pAD7 (Key resources table) was used to overexpress NeRh50 in the *E. coli* strain GT1000 (*Javelle et al., 2004*). GT1000 was transformed with pAD7 and grown in M9 medium (*Elbing and Brent, 2002*), in which ammonium was replaced by 200 µg/ml glutamine as sole nitrogen source. NeRh50 was purified as described by *Lupo et al., 2007* with minor modifications, namely: the membrane was solubilized using 2% lauryldecylamine oxide (LDAO) instead of 5% *n*-octyl-β-D-glucopyranoside (OG), and 0.09% LDAO was used in place of 0.5% β-OG in the final size exclusion chromatography buffer (50 mL Tris pH 7.8, 100 mL NaCl, 0.09% LDAO).

## AmtB and NeRh50 insertion into proteoliposomes

AmtB and NeRh50 were inserted into liposomes containing *E. coli* polar lipids/phosphatidylcholine (POPC) 2/1(wt/wt) as previously described (*Mirandela et al., 2019*). For each AmtB variant, proteoliposomes were prepared at lipid-to-protein ratios (LPRs) of 5, 10, and 50 (wt/wt). The size distribution of proteoliposomes was measured by dynamic light scattering (DLS) using a Zetasizer Nano ZS (Malvern Instruments, Malvern, UK). This analysis showed that the proteoliposomes had an average diameter of 110 nm (*Figure 8*). Proteoliposomes were divided into 100 µL aliquots and stored at −80°C.

To ensure that all AmtB variants were correctly inserted into the proteoliposomes, the proteoliposomes were solubilized in 2% DDM and the proteins analyzed by size exclusion chromatography using a superdex 200 (10 × 300) enhanced column. The elution profile of all variants and the wild-type were identical, showing a single monodisperse peak eluting between 10.4–10.6 ml (*Figure 4— figure supplement 1*). This demonstrated that all proteins were correctly folded, as trimers, in the proteoliposomes.

## Solid supported membrane electrophysiology

To form the solid-supported membrane, 3 mm gold-plated sensors were prepared according to the manufacturer's instructions (Nanion Technologies, Munich, Germany), as described previously (*Bazzone et al., 2017*). Proteoliposomes/empty liposomes were defrosted and sonicated in a sonication bath at 35 W for 1 min, diluted 10 times in non-activating (NA) solution (*Supplementary file 1*), and then 10 µl were added at the surface of the SSM on the sensor. After centrifugation, the sensors were stored at 4°C for a maximum of 48 hr before electrophysiological measurements. For the $D_2O$ experiments, all the solutions were prepared using $D_2O$ instead of water.

All measurements were made at room temperature (21°C) using a SURFE$^2$R N1 apparatus (Nanion Technologies, Munich, Germany) with default parameters (*Bazzone et al., 2017*). Prior to any measurements, the quality of the sensors was determined by measuring their capacitance (15–30 nF) and conductance (<5 nS).

For functional measurements at a fixed pH, a single solution exchange protocol was used with each phase lasting 1 s (*Bazzone et al., 2017*). First, non-active (NA) solution was injected onto the sensor, followed by activating (A) solution containing the substrate at the desired concentration and finally NA solution (*Supplementary file 1*).

For the measurements under inwardly orientated pH gradient, a double solution exchange protocol was used (*Bazzone et al., 2017*), in which an additional resting solution phase of 15 min in NA solution at pH 8 was added to the end. The incubation phase adjusts the inner pH of the proteoliposomes to

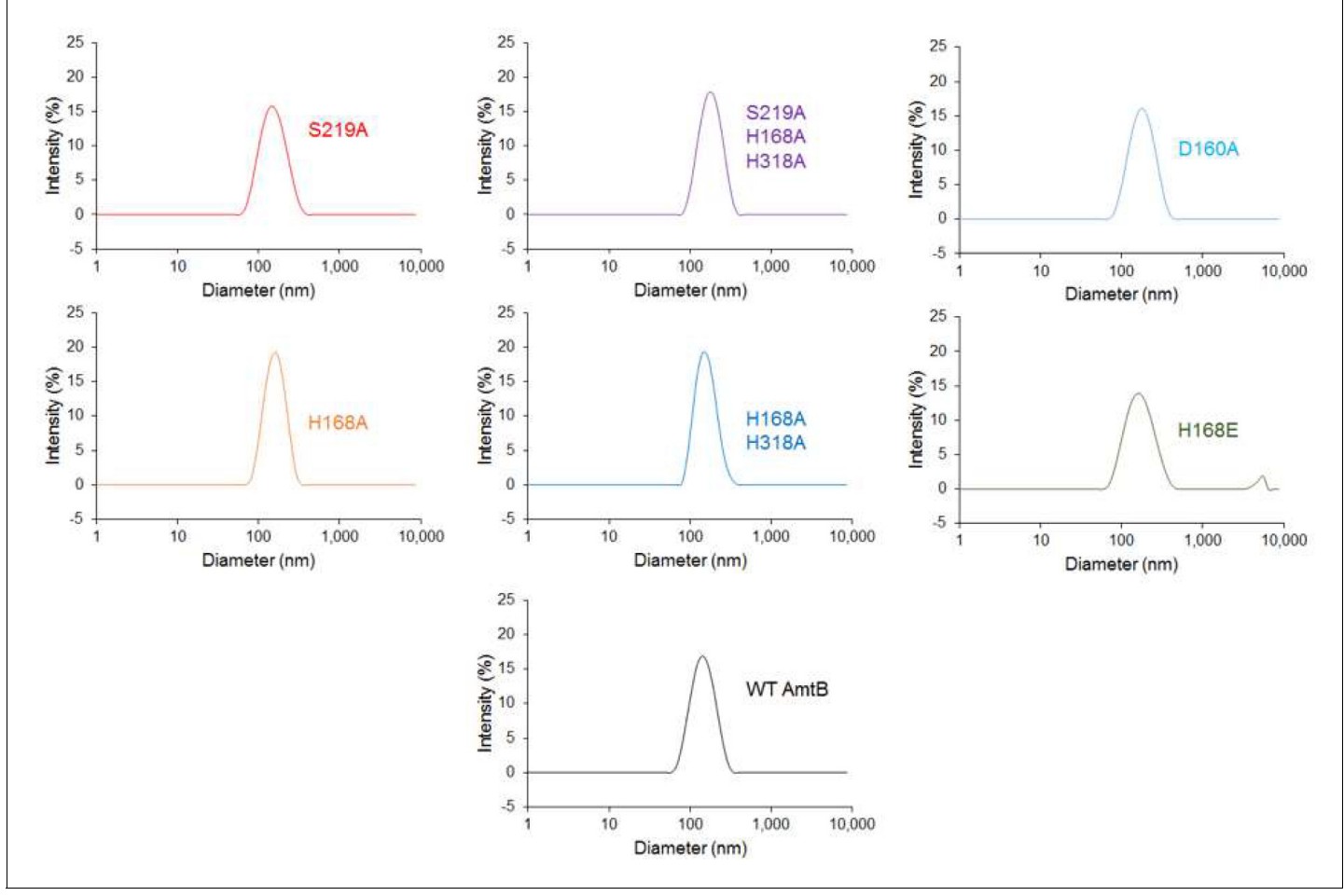

**Figure 8.** Size distribution of the proteoliposomes containing wild-type and variants of AmtB. Dynamic light scattering was used to determine the number-weighted distribution of liposome sizes in the detection reagent. The distribution was monodisperse, with a mean diameter of 110 nm.

pH 8 and establishes a pH gradient at the beginning of each measurement by pulsing the activation solution at pH 5.

Each sensor was measured in the order pH (in/out) 7/7, 8/5 (with $NH_4^+$), 8/5 (without $NH_4^+$), and finally again 7/7 to ensure that the signals do not significantly decrease with time. The data are normalized against the measurements conducted at pH7 in/out for each sensor. All measurements were recorded on 8 sensors from two independent protein purification batches, with 3 measurements recorded for each sensor.

The kinetic parameters were calculated using Graphpad Prism 6 (GraphPad Software, San Diego, California, USA) and fitted according to the Michaelis-Menten equation (Key resources table). Lifetime analysis of the current (decay time of the transient current) was performed to differentiate small pre-steady state currents, which arise due to the binding of a charged species to membrane proteins, from currents reflecting full transport cycles, which show faster decay rates under raised liposomal LPR (*Bazzone et al., 2017*). The decay time of the transient current (*Table 1*) was calculated by fitting the raw transient current data between the apex of the peak and the baseline (after transport) with a non-linear regression using OriginPro 2017 (OriginLab, Northampton, Massachusetts, USA). The regression was done using a one-phase exponential decay function with time constant parameter:

$$y = y_0 = A_1 e^{-x/t_1}$$

The fit was obtained using the Levenberg-Marquardt iteration algorithm, where $x$ and $y$ represent coordinates on the respective axis, $y_0$ represents the offset at a given point, $A$ represents the amplitude, and $t$ is the time constant.

## Molecular Dynamics simulations

The AmtB trimer (PDB code: 1U7G) (*Khademi et al., 2004*) was processed using the CHARMM-GUI web server (*Lee et al., 2016*). Any mutations inserted during the crystallization process were reverted to the wild-type form. The N-termini and C-termini of the subunits were capped with acetyl and N-methyl amide moieties, respectively. The protein was then inserted into a membrane patch of *xy*-dimensions 13 × 13 nm. Unless otherwise specified, a membrane composition of palmitoyl oleoyl phosphatidyl ethanolamine and palmitoyl oleoyl phosphatidyl glycine (POPE/POPG) at a 3:1 ratio was used in order to approximate the composition of a bacterial cytoplasmic membrane. We employed the CHARMM36 forcefield for the protein and counter ions (*Best et al., 2012*). The water molecules were modeled with the TIP3P model (*Jorgensen et al., 1983*). Water bonds and distances were constrained by the Settle method (*Miyamoto and Kollman, 1992*), and all other bonds by the LINCS method (*Hess et al., 1997*). In simulations without ammonium, $K^+$ and $Cl^-$ ions were added to neutralize the system and obtain a bulk ionic concentration of 250 mM. In simulations with ammonium, $K^+$ was replaced by $NH_4^+$. The parameters for $NH_4^+$ and $NH_3$ (umbrella sampling simulations) were adapted from *Nygaard et al., 2006*.

After a steepest descent energy minimization, the system was equilibrated by six consecutive equilibration steps using position restraints on heavy atoms of 1000 $kJ/mol.nm^2$. The first three equilibration steps were conducted in an NVT ensemble, applying a Berendsen thermostat (*Berendsen et al., 1984*) to keep the temperature at 310K. The subsequent steps were conducted under an NPT ensemble, using a Berendsen barostat (*Berendsen et al., 1984*) to keep the pressure at 1 bar. Production molecular dynamics simulations were carried out using a v-rescale thermostat (*Bussi et al., 2007*) with a time constant of 0.2 ps, and a Berendsen barostat with semi-isotropic coupling. A timestep of 2 fs was used throughout the simulations.

In a subset of simulations, we aimed to test the effect of membrane voltage on the internal hydration of AmtB using CompEL. For the CompEL simulations (*Kutzner et al., 2016*), the system was duplicated along the z-axis, perpendicular to the membrane surface. To focus on the physiologically relevant voltage gradient in *E. coli*, that is a negative potential on the inside of the cell of magnitude −140 to −170 mV (*Cohen and Venkatachalam, 2014*), an antiparallel orientation of the two trimers in the double bilayers was used (*Felle et al., 1980*). The final double system consisted of a rectangular box of 13 × 13×20 nm. For the CompEL simulations, 1000 positively charged (either $NH_4^+$ or $K^+$) and 1000 negatively charged ions ($Cl^-$) were added to the system, then the system was neutralized, and the desired ion imbalance established.

The Umbrella Sampling (US) Potential-of-Mean-Force (PMF) calculations (*Torrie and Valleau, 1977*) were set up as described previously by *Hub et al., 2010b*. A snapshot was taken from the simulation of the single bilayer system with the twin-His motif in the DE protonation state and both the CWW and PWW occupied. The pore coordinates were obtained using the software HOLE (*Smart et al., 1996*), removing the solvent and mutating F215 to alanine during the HOLE run only. Starting coordinates for each umbrella window were generated by placing $NH_3$ in the central x-y coordinate of the pore defined by HOLE at positions every 0.5 Å in the z coordinate. Solvent molecules within 2 Å of the ammonia's N atom were removed. Minimization and equilibration were then re-performed as described above. Unless otherwise stated, position restraints were used for all water oxygen atoms in the CWW with a 200 $kJ/mol.nm^2$ force constant; while the TIP3 molecules within the lower water wire were not restrained. For the US the N atom of ammonia was position-restrained with a force constant of 1000 $kJ/mol.nm^2$ on the z axis and a 400 $kJ/mol.nm^2$ cylindrical flat-bottomed potential with a radius of 5 Å in the x-y plane, as described earlier by *Hub et al., 2010a*. For some US window simulations, the ammonia z-axis restraints were increased and the time step reduced in the equilibration to relax steric clashes between sidechains and ammonia. After equilibration, US simulations were run for 10ns, using the parameters described above (*Lee et al., 2016*) and removing the initial two ns for further equilibration. The PMF profiles were generated with the GROMACS implementation of the weighted histogram analysis method (WHAM) with the periodic implementation (*Hub et al., 2010a*). Further US simulations were performed to as needed to improve sampling in regions of the profile that were not sufficiently sampled. The Bayesian bootstrap method was performed with 200 runs to calculate the standard deviation of the PMF.

## Free energy calculations for proton translocation

The free energies for proton translocation were evaluated by protonating the water molecules at different sites along the periplasmic and cytoplasmic water wires. Electrostatic effects in proteins are often treated more effectively using semi-macroscopic models which can overcome the convergence problems of more rigorous microscopic models. Here we used the semi-macroscopic protein dipole/Langevin dipole approach of Warshel and coworkers in the linear response approximation version (PDLD/S-LRA) (*Kato et al., 2006*; *Sham et al., 2000*). Positions of the water molecules in the PWW and CWW were obtained from the corresponding MD snapshots (*Figure 1*). All PDLD/S-LRA p$K_a$ calculations were performed using the automated procedure in the MOLARIS simulations package (*Lee et al., 1993*) in combination with the ENZYMIX force field. The simulation included the use of the surface-constrained all atom solvent model (SCAAS) (*Warshel and King, 1985*) and the local reaction field (LRF) long-range treatment of electrostatics. At each site, 20 configurations for the charged and uncharged state were generated. The obtained p$K_a$ values were then converted to free energies for proton translocation.

## Acknowledgements

Special thanks to Prof. Iain Hunter (Strathclyde Institute of Pharmacy and Biomedical Sciences) for invaluable discussions and help during this project. We also thank Pascale Van Vooren for technical support and Thomas P Jahn for sharing the *triple-mepΔ trk,1,2Δ* yeast strain. Anna Maria Marini is a senior research associate FNRS and WELBIO investigator and thanks the help of FRFS-WELBIO grant ref: CR-2019A-05R.

## Additional information

### Funding

| Funder | Grant reference number | Author |
| --- | --- | --- |
| Tenovus | S17-07 | Arnaud Javelle |
| Scottish Universities Physics Alliance | | Ulrich Zachariae |
| Natural Environment Research Council | NE/M001415/1 | Paul A Hoskisson |
| Fonds De La Recherche Scientifique - FNRS | WELBIO grant ref: CR-2019A-05R. | Anna-Maria Marini |

The funders had no role in study design, data collection and interpretation, or the decision to submit the work for publication.

### Author contributions

Gordon Williamson, Data curation, Formal analysis, Investigation, Methodology, Writing - review and editing; Giulia Tamburrino, Adriana Bizior, Data curation, Formal analysis, Investigation, Writing - review and editing; Mélanie Boeckstaens, Conceptualization, Data curation, Formal analysis, Methodology, Writing - review and editing; Gaëtan Dias Mirandela, Callum M Ives, Data curation, Formal analysis, Investigation; Marcus G Bage, Formal analysis, Investigation; Andrei Pisliakov, Conceptualization, Formal analysis, Methodology, Writing - review and editing; Eilidh Terras, Investigation; Paul A Hoskisson, Formal analysis, Funding acquisition, Writing - review and editing; Anna Maria Marini, Conceptualization, Methodology, Project administration, Writing - review and editing; Ulrich Zachariae, Conceptualization, Formal analysis, Methodology, Writing - original draft, Project administration; Arnaud Javelle, Conceptualization, Data curation, Formal analysis, Supervision, Investigation, Methodology, Writing - original draft, Project administration

### Author ORCIDs

Gordon Williamson (iD) https://orcid.org/0000-0003-3053-8322
Mélanie Boeckstaens (iD) https://orcid.org/0000-0003-1629-7403

Gaëtan Dias Mirandela [ID] https://orcid.org/0000-0001-5871-6288
Andrei Pisliakov [ID] https://orcid.org/0000-0003-1536-0589
Callum M Ives [ID] http://orcid.org/0000-0003-0511-1220
Arnaud Javelle [ID] https://orcid.org/0000-0002-3611-5737

Decision letter and Author response
Decision letter https://doi.org/10.7554/eLife.57183.sa1
Author response https://doi.org/10.7554/eLife.57183.sa2

## Additional files

**Supplementary files**
- Supplementary file 1. Supplementary Table 1.
- Transparent reporting form

### Data availability

All data generated or analysed during this study are included in the manuscript and supporting files. Source data files have been provided for Figures 1-5 and Table 2. Simulation code is available on GitHub at https://github.com/UZgroup/A-two-lane-mechanism-for-selective-biological-ammonium-transport/ (copy archived at https://github.com/elifesciences-publications/A-two-lane-mechanism-for-selective-biological-ammonium-transport) and the trajectory files are available on Figshare (https://doi.org/10.6084/m9.figshare.12826316).

The following dataset was generated:

| Author(s) | Year | Dataset title | Dataset URL | Database and Identifier |
|---|---|---|---|---|
| Tamburrino G, Zachariae U | 2020 | Molecular dynamics simulation trajectories, AmtB in twin-His HSD-HSE and HSE-HSD states | https://figshare.com/articles/dataset/Molecular_dynamics_simulation_trajectories_AmtB_in_twin-His_HSD-HSE_and_HSE-HSD_states/12826316 | figshare, 10.6084/m9.figshare.12826316 |

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
