## [Decision Letter]

**Acceptance summary:**

A radically new mechanism is suggested here for the transport of NH_4_^+^ across the cytoplasmic membrane. According to this suggestion, strongly supported by molecular simulations, as well as various biophysical, mutagenesis, and cellular assays, NH_4_^+^ transport involves deprotonation, translocation of NH_3_ via hydrophobic pathway and the proton via two water-wires connected to each other by two Histidines, and reprotonation in the other end.

**Decision letter after peer review:**

Thank you for submitting your article "A two-lane mechanism for selective biological ammonium transport" for consideration by *eLife*. Your article has been reviewed by three peer reviewers, including Nir Ben-Tal as the Reviewing Editor and Reviewer #1, and the evaluation has been overseen by Olga Boudker as the Senior Editor. The following individuals involved in review of your submission have agreed to reveal their identity: Oded Lewinson (Reviewer #2); Ian Collinson (Reviewer #3).

The reviewers have discussed the reviews with one another and the Reviewing Editor has drafted this decision to help you prepare a revised submission.

Summary:

Nature have 'invented' various mechanisms to meet the challenge of selective translocation of charged molecules across the membrane. A study of two homologous NH_4_^+^ transporters, *Escherichia coli* AmtB and *Nitrosomonas europaea* Rh50, presented in this manuscript, suggests a new and very unique mechanism. According to this suggestion, strongly supported by molecular simulations, as well as various biophysical, mutagenesis, and cellular assays, NH_4_^+^ transport involves deprotonation, translocation of NH_3_ via hydrophobic pathway and the proton via two water-wires connected to each other by two Histidines, and reprotonation in the other end. Key amino acids that facilitate both proton and NH_3_ transfer are highlighted.

This is a very well thought of and carefully conducted study, and the proposed mechanism is revolutionary. The manuscript can be published after the authors will address a few issues.

Essential revisions:

1) The suggested pathways emerged from umbrella sampling simulations, which require predetermination of the "reaction coordinate". How certain the authors are about the proper choice of these for the water wires and NH_3_ translocation pathways? Maybe consider using methods like metadynamics that may (or may not) detect the pathways without prejudice? Else, at the very least, this point, i.e., the arbitrarily chosen pathways, should be discussed.

2) The data in Figure 3 is normalized, but the authors do not explain how or why. It would be interesting to know whether there is any pH effect on transport. Presumably the initial deprotonation event would be strongly inhibited by lowering the external pH? So if there is no effect, could it mean that the deprotonation event is very fast? Or something else?

3) It is important to show decay at different lipid-to-protein ratios in all figures.

4) "The K^+^ translocation activity is not saturable in the substrate range [12.5-200 mM]".

Why were higher values not tested? This would help convey the specificity message. In addition, this suggested insatiable substrate-dependence should be compare to the proton affinity.

5) "On the cytoplasmic face, the pH of the cell interior leads to recombination to NH_4_^+^"

Why is that? The cell's interior should be slightly basic.

6) "Summarizing, these results suggest that AmtB switches from transporter- to channel-like activity in the absence of the twin-His motif, directly translocating hydrated NH_4_^+^ through the pore." If we understand correctly, this conclusion is drawn from the data of the mutants in which case it may be irrelevant for the WT.

7) "Additionally, it was impossible to determine with confidence a catalytic constant (Km) for both variants since no clear saturation was reached, even after an ammonium pulse of 200 mM (Figure 4B)." The D160A data in Figure 4B looks like a reasonable Michaelis Menten fit. We suggest using higher ammonium concentration (can go up to 800 mM in SSME) and report data and fits.

8) In Figure 4C, the activity of the WT should be shown to allow for comparisons.

9) For reproducibility the authors should make the simulations readily available to the public. All the relevant parameter files, trajectory, etc.

---

## [Author Response]

Essential revisions:1) The suggested pathways emerged from umbrella sampling simulations, which require predetermination of the "reaction coordinate". How certain the authors are about the proper choice of these for the water wires and NH_3_ translocation pathways? Maybe consider using methods like metadynamics that may (or may not) detect the pathways without prejudice? Else, at the very least, this point, i.e., the arbitrarily chosen pathways, should be discussed.

We thank the reviewers for this question and for giving us the opportunity to clarify this point.

The transport mechanism we propose rests on two key pathways – the proton transfer pathway and an independent pathway for neutral ammonia. Previously, before evidence for electrogenic transport was obtained by us and other groups, the consensus in the field had favoured transport of neutral NH_3_ only, along a hydrophobic route (shown, e.g. by simulations on Rh50 [Hub et al., 2010]) similar to the one we find for AmtB.

Importantly, our novel finding of a separate transfer pathway for proton transport, the missing link to explain electrogenic transport and the central new element of the proposed mechanism, was not obtained from umbrella sampling simulations along a predetermined pathway. Instead, we used classical simulations, in which two water wires formed spontaneously. The location of these water molecules as well as the conformations of residues lining the pathway, come from unbiased molecular dynamics simulations. These spontaneously formed configurations were then subsequently used to calculate energy barriers for proton transport.

The separate pathway for NH_3_ translocation was indeed probed using umbrella sampling, however, we restrained only the z-position along the pore axis of NH_3_. This way, the molecule was free to move in x and y and thus to find its optimal position within the pore, in particular as we used extended sampling times. The starting points were optimised following detection of the most likely translocation pathway through the protein by using the program HOLE. Moreover, the pathway and energy profiles we identify for AmtB and agree with the results for Rh50 by Hub et al., 2010. All of these points give us a great level of confidence over the validity of both pathways.

Metadynamics simulations are not completely devoid of an element of choice, as for example collective coordinates have to be defined. Comparing to metadynamics, we therefore trust that our approach provides a similar, or even higher level of freedom for the particle to find its optimal pathway.

Following a suggestion from the reviewers, we have now added a discussion on the choice of initial positions we used in umbrella sampling simulations.

2) The data in Figure 3 is normalized, but the authors do not explain how or why. It would be interesting to know whether there is any pH effect on transport. Presumably the initial deprotonation event would be strongly inhibited by lowering the external pH? So if there is no effect, could it mean that the deprotonation event is very fast? Or something else?

The data are normalised against the measurements done at pH7 in/out, and the figure legends and text in the Materials and methods section have now been amended to make this clearer.

We have done this because we always measured the sensor readouts in the order pH (in/out) 7/7, 8/5, 8/5 (this time without NH_4_^+^), and finally 7/7 again to be sure that the signals do not significantly decrease with time. We also mentioned this procedure in the Materials and methods section: “The activity of AmtB and NeRh50 was tested at pH 7 before and after each measurement to ensure that there was no activity loss during the measurements.” We therefore normalised all the measurements of the series against the measurement at pH 7/7 of the corresponding series.

We have measured the effect of various static pH values on transport (pH in/out, 5/5, 6/6, 7/7, 8/8), and indeed observed a reduction of the maximum amplitude by a factor of 2 between pH 7 and 5 (see Author response image 1), as was also observed for another Amt from *A. fulgitus* (see Wacker et al., 2014). However, we think that it is difficult to interpret this phenomenon with confidence and to directly link the pH effect on the deprotonation or any specific step in the translocation process. When changing the pH it is not clear if only the local pH at the deprotonation site is varied accordingly, or if the whole protein is affected. Therefore, to ensure that we do not observe a general effect of the pH on the protein activity, in the experiment presented in figure 3, we pulsed the substrate at pH 5 for 1 second and re-measured thereafter at pH 7.

In the present manuscript, we believe that showing the data measured at various static pH values will not add important new information and may be confusing to readers. We think that a more in-depth analysis of the pH effect would be required to correctly analyse the data and we would prefer to reserve this for a future paper (see, e.g., Bazzone et al., PlosOne, 2016). However, if it is the opinion of the editor and reviewers that we should present these data, we are happy to do so. Please see Author response image 1, the data for the effect of static pH for AmtB and NeRh50 (maximum amplitude and decay time).

3) It is important to show decay at different lipid-to-protein ratios in all figures.

We currently show all the decay times at various LPR in Table 1. The values represent the means of measurements recorded on 8 sensors from two independent protein purification batches. The raw data are given in the source file. In our opinion, showing all the traces within the main text figures will make the figures too busy and confusing. We therefore decided to provide traces at different LPR in two new figures: Figure 1—figure supplement 1 and Figure 5—figure supplement 1.

4) "The K^+^ translocation activity is not saturable in the substrate range [12.5-200 mM]".Why were higher values not tested? This would help convey the specificity message. In addition, this suggested insatiable substrate-dependence should be compare to the proton affinity.

In the experimental assays, the total ionic concentration is maintained at 300 mM (see Supplementary file 1). To measure the activity using K^+^ concentrations higher than 200 mM, we would have to significantly increase the total ionic concentration to maintain the electrostatic balance when applying the substrate pulse. However, it has been shown that increasing the salt concentration is likely to introduce artefacts when recording the signal, which are difficult to control (see Bazzone et al., 2017). We in fact tried to pulse with higher K^+^ or NH_4_^+^ concentrations, but an artefact appeared in the signal in the form of an overshoot after the main current, probably due to ion-lipid interactions. At the present time, we are therefore limited to a maximum concentration of about 200 mM by technical constraints.

5) "On the cytoplasmic face, the pH of the cell interior leads to recombination to NH_4_^+^"Why is that? The cell's interior should be slightly basic.

Yes, this is correct. However, the cytoplasmic pH is around 7.2-7.6 (Martinez et al., Appl. Environ. Microbiol. 2012), which is around 2 pH units below the pKa of NH_4_^+^ deprotonation (9.25). Thus at pH 7.5, more than 99% of ammonium is expected to be in the NH_4_^+^ form.

6) "Summarizing, these results suggest that AmtB switches from transporter- to channel-like activity in the absence of the twin-His motif, directly translocating hydrated NH_4_^+^ through the pore." If we understand correctly, this conclusion is drawn from the data of the mutants in which case it may be irrelevant for the WT.

Yes, we agree that it is irrelevant for the WT of AmtB. However, we deem this information to be essential for understanding the mechanism, and especially the role of the twin-His motif in transport selectivity. It also explains why certain variations in the twin-His motif seen in a few species do not ‘apparently’ inactivate the transporter but alter the general mechanistic features. We have rewritten the statement to make these points clearer.

7) "Additionally, it was impossible to determine with confidence a catalytic constant (Km) for both variants since no clear saturation was reached, even after an ammonium pulse of 200 mM (Figure 4B)." The D160A data in Figure 4B looks like a reasonable Michaelis Menten fit. We suggest using higher ammonium concentration (can go up to 800 mM in SSME) and report data and fits.

Unfortunately, due to technical limitations, it is not possible to use a very high ionic concentration (see also our answer to point 4). Briefly, salt concentrations above 300 mM introduce difficult-to-control artefacts when recording the SSME signal (Bazzone et al., 2017), which leads to limitations in the salt range that we can investigate.

8) In Figure 4C, the activity of the WT should be shown to allow for comparisons.

We agree; the figure has been amended accordingly.

9) For reproducibility the authors should make the simulations readily available to the public. All the relevant parameter files, trajectory, etc.

We are more than happy to share all the relevant files with the public. The files of smaller size (parameters etc.) have been uploaded to a publicly accessible GitHub repository:

https://github.com/UZgroup/A-two-lane-mechanism-for-selective-biological-ammonium-transport/

GitHub however has a 300 MB limit, which means we cannot share, even trimmed-down, trajectory files via this route. To share the larger trajectory file, we have uploaded this to Figshare.